# Co-Encapsulation of Multiple Polyphenols in Plant-Based Milks: Formulation, Gastrointestinal Stability, and Bioaccessibility

**DOI:** 10.3390/foods12183432

**Published:** 2023-09-15

**Authors:** Bingjing Zheng, Hualu Zhou, David Julian McClements

**Affiliations:** 1Biopolymers and Colloids Laboratory, Department of Food Science, University of Massachusetts, Amherst, MA 01003, USA; lingbingjing@gmail.com; 2Department of Food Science and Technology, College of Agricultural and Environmental Sciences, University of Georgia, Griffin, GA 30223, USA; hualuzhou@uga.edu; 3Department of Food Science & Bioengineering, Zhejiang Gongshang University, 18 Xuezheng Street, Hangzhou 310018, China

**Keywords:** phytochemicals, curcumin, quercetin, resveratrol, pH-shift, plant-based foods, INFOGEST, in vitro digestion

## Abstract

Plant-based milk is particularly suitable for fortification with multiple nutraceuticals because it contains both hydrophobic and hydrophilic domains that can accommodate molecules with different polarities. In this study, we fortified soymilk with three common polyphenols (curcumin, quercetin, and resveratrol) using three pH-driven approaches. We compared the effectiveness of these three different approaches for co-encapsulating polyphenols. The gastrointestinal fate of the polyphenol-fortified soymilks was then studied by passing them through a simulated mouth, stomach, and small intestine, including the stability and bioaccessibility of polyphenols. All three pH-driven approaches were suitable for co-encapsulating multiple polyphenols at a high encapsulation efficiency, especially for the curcumin and resveratrol. The polyphenol-loaded delivery systems exhibited similar changes in particle size, charge, stability, and bioaccessibility as they passed through the mouth, stomach, and intestinal phases. The bioaccessibility of the co-encapsulated polyphenols was much greater than that of crystallized polyphenols dispersed in water. The poor bioaccessibility of the crystallized polyphenols was attributed to their low solubility in water, which made them more difficult to solubilize within mixed micelles. This study underscores the feasibility of pH-driven approaches for encapsulating a variety of polyphenols into the same plant-based delivery system. These fortified plant-based milks may therefore be designed to provide specific health benefits to consumers.

## 1. Introduction

Polyphenols are a diverse group of natural compounds that are found in numerous edible plants, like fruits, vegetables, tea, and coffee. Researchers are trying to establish the ability of these phytochemicals to improve human health and wellbeing, which is mainly attributed to their antioxidant, antimicrobial, and anti-inflammatory activities [1,2]. Polyphenols are therefore being incorporated into functional foods and beverages to enhance their nutritional profiles [2,3]. However, it is often challenging to incorporate polyphenols into food products. One of the biggest challenges is that lipophilic polyphenols (LPs) have relatively low water solubility, which limits their bioaccessibility, absorption, and effectiveness in the human body [4]. Moreover, they often have a bitter or astringent taste, which adversely impacts the palatability of foods or beverages rich in these polyphenolic substances [5]. For these reasons, specialized formulation technologies are often required to ensure proper dispersibility, stability, flavor profile, bioavailability, and efficacy [4,6,7,8].

The plant-based milk market has been expanding recently for several reasons. Individuals suffering from lactose intolerances or dairy allergies may drink plant-based milks rather than cow’s milk. Individuals concerned about animal welfare or about the adverse effects of the livestock industry on environmental sustainability may also drink plant-based milks [9]. Many kinds of plant-based milks are available commercially, including almond, soybean, oat, coconut, and rice milks. These products are usually colloidal dispersions that consist of oil droplets or oil bodies dispersed within an aqueous medium, which makes them particularly suitable for fortification with multiple nutraceuticals [10]. For example, the hydrophobic interior of oil droplets can solubilize lipophilic polyphenols, whereas the surrounding aqueous phase can solubilize hydrophilic ones. Moreover, some plant proteins in these products can bind hydrophobic polyphenols and protect them from physical or chemical degradation. Plant-based milks therefore offer a particularly versatile vehicle for encapsulating, protecting, and delivering multiple polyphenols [11].

Lipophilic polyphenols are typically isolated from plants and converted into powders, which can lead to difficulties in dispersing them in plant-based milks because they are prone to clumping, sedimentation, and uneven distribution [1,7,12]. Consequently, there is a need for economically viable methods that can be used to incorporate multiple polyphenols into plant-based milks. pH-driven methods have been used for the encapsulation of polyphenols into bovine milk and plant-based milks [13,14,15]. These methods are based on the fact that polyphenols tend to be highly soluble under alkaline conditions but insoluble under acidic ones. Thus, the polyphenols can be solubilized in an alkaline solution that is then mixed with a slightly acidic colloidal system that contains particles with a hydrophobic core [16,17]. Consequently, the polyphenols move from the hydrophilic aqueous phase to the hydrophobic core of the oil droplets when the pH is changed from basic to acidic [18,19,20,21,22,23]. 

Recently, pH-driven approaches were successfully used to encapsulate lipophilic polyphenols into plant-based milk (such as soy, almond, cashew, coconut, or oat milk) [24]. First, the polyphenols were dissolved in an alkaline solution (pH 12), which was then mixed with a plant-based milk, thereby causing the polyphenols to move from the water into the hydrophobic core of the oil droplets [25]. The initial and final pH of the plant-based milk usually has to be controlled to ensure that encapsulation occurs, without inducing precipitation or flocculation of other components in the system [26]. A recent study showed that curcumin could be introduced into whole, low-fat, and skim milks using a pH-driven approach, which increased its in vitro bioaccessibility and in vivo antioxidant activity [27]. In the current study, we aim to determine whether pH-driven approaches can be used to encapsulate multiple polyphenols in the same plant-based milk. 

We selected soymilk as a model plant-based milk and curcumin, quercetin, and resveratrol as model polyphenols. Soymilk contains oil bodies with a hydrophobic interior, which should be suitable for solubilizing lipophilic polyphenols [28]. Curcumin, resveratrol, and quercetin were used as model polyphenols because they have been commonly used as model nutraceuticals in previous studies and are representatives of flavonoids and non-flavonoids with different molecular characteristics. A unique aspect of this study is that we compare three different formulation protocols for loading multiple polyphenols into plant-based milk. We hypothesize that each of these protocols has its own advantages and disadvantages that make it suitable for specific applications. 

Protocol 1: All three powdered polyphenols were dissolved in the same alkaline solution, which was then mixed with the soymilk ((CQR)M). This approach has the advantage that multiple polyphenols can be simultaneously encapsulated into the soymilk, thereby simplifying the process;Protocol 2: Each powdered polyphenol was dissolved in a different alkaline solution, which was then mixed with the soymilk ((C+Q+R)M)). This approach may be useful in situations where different polyphenols require different pH conditions to be fully dissolved and remain stable;Protocol 3: Each powdered polyphenols was dissolved in a different alkaline solution and then loaded into soymilk individually, and then the three different polyphenol-loaded soymilks were combined (CM+QM+RM). This approach may be useful for controlling the ratio of different polyphenols required in a soymilk product, which could be advantageous for personalized nutrition applications.

We then investigated the gastrointestinal stability and bioaccessibility of the three polyphenols loaded into the soymilk using the three different pH-driven methods. We hypothesized that the preparation method used may impact the stability and transfer of the polyphenols in the system, which could impact their gastrointestinal fate. This study should provide important insights into the most suitable methods for fortifying plant-based milks with multiple polyphenols. 

## 2. Materials and Methods

### 2.1. Materials

Plant-based milk (Silk Vanilla Soy Creamer, 32 oz) was obtained from a local grocery store, which was reported to contain 10 wt% oil. Curcumin (C2302), resveratrol (R0071), and quercetin (P0042) were obtained from TCI America (Portland, OR, USA), which were reported to have purities exceeding 97.0%, 99.0%, and 96.0%, respectively. The chemicals employed in the in vitro digestion model included inorganic salts, porcine mucin (M2378, Type II), porcine pepsin (P7000, ≥250 units/mg solid), porcine pancreatin (P7545), porcine lipase (L3126, Type II), and porcine bile extract (B8631), which were purchased from Sigma-Aldrich Chemical Co. (St. Louis, MO, USA). The organic solvents used, ethanol and acetonitrile, were of HPLC grade, and all solutions and buffers were prepared using double distilled water.

### 2.2. Formulations of Multiple Polyphenols Fortified Soymilk

We incorporated three types of LPs (curcumin, quercetin, or resveratrol) sequentially into soymilk, following established protocols as outlined in our previous study [29]. In brief, we categorized our approach into four distinct groups:Crystalline: Crystalline curcumin (C), quercetin (Q), and resveratrol (R) were dispersed in water.(C+Q+R)M: Crystalline curcumin (C), quercetin (Q), and resveratrol (R) were individually dissolved in separate alkaline solutions that were then introduced into the soymilk. A NaOH solution of pH 12 was used to prepare all samples in this study.(CQR)M: The three nutraceuticals were all dissolved in the same alkaline solution before being introduced into the soymilk.CM+QM+RM: Each of the three nutraceuticals was dissolved in a separate alkaline solution and separately loaded into soymilk before the three soymilks were combined together.

Finally, all the groups had the same oil concertation (5 wt%) and polyphenol concentration (2 mg/g oil for curcumin, resveratrol, and quercetin). Then, they were stored in a refrigerator (4 °C) before carrying out the following analyses. 

### 2.3. Color and Appearance

The color coordinates of the samples were measured using an instrumental colorimeter (ColorFlex EZ 45/0 LAV, Hunter Colorimeter, Hunter, VA, USA). Tristimulus color coordinates (L*, a*, b* values) were measured for each sample. Standardized black and white plates were utilized for instrument calibration. Measurements were performed using a standardized light source (D65) and detection angle (10°). Samples were poured into transparent Petri dishes and covered with a black cup during analysis. The appearances of the samples were also characterized by taking photographs using a digital camera.

### 2.4. In Vitro Gastrointestinal Model 

A standardized INFOGEST model was used to investigate the gastrointestinal behavior of crystalline LP dispersions and LP-fortified plant-based milks [30]. Simulated gastrointestinal fluids were prepared according to the INFOGEST guidelines, including the analysis and standardization of enzyme activities. The details of the method used are given in our previous studies [31]. For this reason, only a concise overview of the method is given here.

Mouth Phase: LP-fortified plant-based milks (5 g, 5 wt% oil) were mixed with simulated saliva (5 mL) to simulate oral processing. The simulated saliva contained inorganic salts and 3 mg/mL porcine mucin. The pH value of this solution was then adjusted to 7.0.

Stomach Phase: At the end of the mouth phase, a sample was collected (10 mL) and then combined with simulated gastric fluids (10 mL), adjusted to pH 3, and rotated for 2 h to replicate gastric conditions. The simulated gastric fluid contained HCl, salts, and pepsin (~2000 U/mL).

Small Intestine Phase: At the end of the stomach phase, a sample was collected (20 mL) and mixed with simulated intestinal fluids (20 mL), and then the mixture was adjusted to pH 7. The simulated small intestine fluids contained inorganic salts, bile salts, pancreatin, and lipase (10 mM bile salts, lipase activity 2000 U/mL). After 2 h of incubation, the samples were centrifuged at 4 °C (18,000 rpm) and the clear “mixed micelle” layer was collected.

### 2.5. Particle Dimensions and Charge

The mean particle diameter of the LP-fortified plant-based milks was assessed before and after gastrointestinal passage using a laser diffraction device (Mastersizer 2000, Malvern Instruments, Worcestershire, UK). For surface charge determination, the particle electrophoresis mode of the Zetasizer instrument (Nano-ZS, Malvern Instruments) was used. Samples were diluted with water adjusted to the same pH as them before being analyzed to obtain a sufficiently strong light scattering signal.

### 2.6. Encapsulation Efficiency, GIT Stability, and Bioaccessibility of Polyphenols

High-performance liquid chromatography (HPLC, Agilent 1100 series, Agilent Technologies, Santa Clara, CA, USA) with a UV-visible detector was used to quantify the concentrations of the three different polyphenols in the samples before and after digestion. First, the LPs were extracted from the samples using an ethanol solution containing 1% acetic acid. Then, they were quantified by measuring the UV-visible absorbance at a wavelength appropriate for each polyphenol: 307 nm (resveratrol), 420 nm (curcumin), and 370 nm (quercetin). The mobile phase composition was also polyphenol-specific, with acetonitrile and 1% acetic acid in ratios of 30:70, 55:45, and 30:70 for resveratrol, curcumin, and quercetin, respectively. The R^2^ values for the absorbance–concentration standard curves for each polyphenol exceeded 0.999.

The measured concentrations of the LPs in each gastrointestinal phase were then used to calculate the encapsulation efficiency (EE), gastrointestinal tract (GIT) stability, and bioaccessibility [31]:
(1)$$EE\;\left(\%\right)=100\times \frac{\;C_{NE}\;}{C_{0}},$$
(2)$$GIT\;stability\;\left(\%\right)=100\times \frac{\;C_{D}*8}{C_{NE}},$$
(3)$$Bioaccessibility\;\left(\%\right)=100\times \frac{\;C_{M}\;}{C_{D}}.$$
where C_NE_ is the measured LP concentration in the soymilk after preparation, while C_0_ is the actual LP concentration used to prepare the soymilk. C_D_ is the LP concentration found in the digesta, and C_M_ is the LP concentration found in the mixed micelle phase. The factor “8” appears in Equation 3 to account for sample dilution as the samples pass through the gastrointestinal model. Specifically, the sample volume transitions from an initial 5 mL to a final 40 mL in the small intestine. To ensure the encapsulation of LPs in the delivery system, relatively mild centrifugation conditions (5000 rpm, 10 min, 25 °C) were used to remove non-encapsulated LPs. More severe centrifugation conditions might have destabilized the soymilk. It should be noted that the bioaccessibility provides insights into the fraction of LPs in the small intestine that are solubilized in mixed micelles and therefore available for absorption, whereas the GIT stability provides information about the fraction of LPs that survive oral and stomach conditions and therefore reach the small intestinal intact. Consequently, the overall bioaccessibility should be the product of the bioaccessibility and GIT stability.

### 2.7. Statistical Analysis

For each experimental treatment, a minimum of two samples were freshly prepared. For each sample, three repeat measurements were made. The mean and standard deviation were then calculated from these values. To determine significant differences among the samples, an ANOVA approach was employed, followed by a post hoc Tukey HSD test with a significance threshold of *p* < 0.05.

## 3. Results and Discussion

### 3.1. Color and Appearance of Polyphenol-Fortified Soymilk

Initially, curcumin, resveratrol, and quercetin were loaded into the soymilk using the pH shift methods. For all soymilk samples, no sedimentation of the polyphenols was observed, or evidence of phase separation of the soymilk (Figure 1). All fortified soymilk products appeared yellow in color, which can mainly be attributed to the presence of curcumin. The appearances of the polyphenol-fortified soymilks prepared using the different pH-driven approaches were similar, but the crystalline polyphenols dispersed in the pure water system appeared to be insoluble and to precipitate. Presumably, the oil bodies in the soymilk were able to incorporate the non-polar polyphenols into their hydrophobic interiors (Figure 1b). These results suggest that soymilk is a suitable food product for simultaneously encapsulating curcumin, resveratrol, and quercetin. 

Interestingly, the method of incorporating the polyphenols into the soymilk did appear to alter its appearance. The (C+Q+R)M sample had a higher yellowness (b*) than the other two samples. Dissolving the powdered polyphenols individually into the alkaline solution may have led to a stronger color because it took less time to dissolve the curcumin powder, and so there was less curcumin degradation.

### 3.2. Encapsulation Efficiency

Ideally, it is important to important to encapsulate a high percentage of polyphenols into soymilk products, as this will increase the color intensity and biological activity. HPLC analysis indicated that all three pH-driven methods successfully loaded all three polyphenols into the same soymilk carrier. However, curcumin and resveratrol had a significantly higher encapsulation efficiency than quercetin in the soymilk products (Figure 2). The samples prepared using the (C+Q+R)M and CM+QM+RM methods have a higher encapsulation efficiency for curcumin and resveratrol, while the (CQR)M method had a slightly better encapsulation efficiency for quercetin than the other methods. These results agree with the color measurements discussed earlier (Figure 1a). Dissolving the three polyphenols into one alkaline solution before adding them to the emulsion, (CQR)M, may have led to a relatively low encapsulation efficiency because it took longer to fully dissolve the powders. As a result, there may have been more chemical degradation of the polyphenols. This result suggests that the time required to dissolve the polyphenol powders should be minimized to achieve a high encapsulation efficiency.

### 3.3. Particle Size and Zeta Potential

After preparation, the fortified soymilk samples prepared using the different pH-driven approaches were passed through the INFOGEST digestion model to determine their gastrointestinal fate. Overall, the method used to prepare the fortified soymilk samples did not have a major impact on their behavior in the simulated gastrointestinal tract (Figure 3). Exposure to the mouth phase had no significant effect on the mean particle sizes or zeta potential values of the soymilk samples. However, exposure to the stomach phase caused a large increase in particle size for all samples. This effect may be due to the highly acidic nature of the gastric fluids, which caused the protein-coated oil bodies to become positively charged, thereby promoting charge neutralization and bridging flocculation by any anionic polysaccharides. Moreover, the protease (pepsin) in the gastric fluids may have partially digested the soymilk proteins, thereby promoting instability [32]. The mean particle diameter increased from below 1 μm to around 60 μm, while the zeta-potential went from highly negative to close to 0 mV, which is consistent with aggregation caused by charge neutralization and bridging effects. The confocal microscopy images indicated extensive protein aggregation (green fluorescence) and oil body aggregation (red fluorescent). These images also showed that the oil droplets and proteins tended to aggregate together. The digestion of the proteins in the coatings around the oil bodies may have promoted some droplet coalescence, leading to large individual oil droplets (Figure 3a) [33]. 

After incubation in the stomach phase, the protein coagulum, oil droplets, and polyphenols are passed into the small intestine phase for further digestion. The small intestine has a higher pH and ionic strength than the stomach. The neutral conditions increase the negative charge on the proteins and protein-coated oil bodies, thereby increasing the electrostatic repulsion between them. As a result, the large aggregates are dissociated, leading to a substantial reduction in particle size. Moreover, the lipids and proteins are hydrolyzed by lipases and proteases, leading to the formation of fatty acids, monoacylglycerols, peptides, and amino acids, which will also reduce the particle size. The fatty acids and monoacylglycerols will then mix with bile salts and form mixed micelles that can solubilize the lipophilic polyphenols.

### 3.4. Gastrointestinal Stability 

The impact of the pH-driven protocols on the gastrointestinal fate of the co-encapsulated polyphenols was also measured (Figure 4). The GIT stability of the polyphenols reflects their ability to resist chemical degradation under gastrointestinal conditions. The concentrations of the polyphenols before and after exposure to different gastrointestinal phases was therefore determined using HPLC. The type of pH-driven protocol used to fortify the soymilk samples had no impact on the polyphenols’ stability (Figure 4). However, the crystalline form of quercetin was much more resistant to chemical degradation during GIT passage than the encapsulated forms. Only around 20–30% of the encapsulated quercetin remained after the digestion process, whereas around 100% of the crystalline quercetin remained. This suggests that the crystalline form of quercetin was much more stable than the solubilized form. There were appreciable differences in the resistance of the different polyphenols to degradation: resveratrol (80–90%) > curcumin (65–70%) > quercetin (20–30%). These differences may have been due to differences in the inherent chemical stability of the polyphenols, as well as due to differences in their location in the system. Polyphenols in hydrophilic regions (like water) are more likely to be more rapidly degraded than those in hydrophobic regions (like oil body or micelle interiors). As discussed in our previous study, the three polyphenols have different oil–water partition coefficients (LogD) [33]: 4.12, 3.39, and 2.17 for curcumin, resveratrol, and quercetin, respectively. The relatively low logD value of quercetin means that a significant fraction (>50%) may be present in the water, thereby facilitating its tendency to precipitate or chemically degrade [33]. It should be noted that the in vitro INFOGEST model cannot accurately mimic all the conditions in the human gastrointestinal tract. Consequently, it will be important to assess whether similar effects are observed using animal or human feeding studies.

### 3.5. Bioaccessibility

The bioaccessibility values of the polyphenols in crystalline form and in soymilk samples prepared using different methods were then measured (Figure 5). Here, the bioaccessibility was taken to be the percentage of a polyphenol in the mixed micelle phase relative to the whole digesta. Overall, the bioaccessibility of the polyphenols encapsulated in the soymilk was significantly higher than for the crystals (Figure 5). In contrast, as discussed earlier, the crystals had very good GIT stability (Figure 4), i.e., they were resistant to chemical degradation. However, their solubilization within the mixed micelles was relatively low, which would be expected to hinder their absorption. Again, the pH-driven method used to load the multiple polyphenols into the soymilk did not have a major impact on their bioaccessibility. This result suggests that once the polyphenols are successfully encapsulated within the soybean oil bodies, they have a similar bioaccessibility. Interestingly, the quercetin had a low GIT stability, but its bioaccessibility was still higher than 50%. The resveratrol and curcumin had an appreciably higher bioaccessibility than the quercetin, which may be because they were more likely to be solubilized within the hydrophobic core of the mixed micelles due to their stronger hydrophobicity. 

## 4. Conclusions

In this study, we investigated the possibility of co-encapsulating three important bioactive polyphenols (curcumin, resveratrol, and quercetin) into a soymilk product. Three different pH-driven encapsulating protocols were compared to determine the most suitable one for this application. Overall, our results indicate that there were no major differences between the three pH-driven methods used, but also that it is important to reduce the amount of time the polyphenols spend under highly alkaline conditions because it may promote their chemical degradation. The simulated gastrointestinal fate of the polyphenol-fortified soymilks was also investigated, including their stability and bioaccessibility. Encapsulation greatly increased the bioaccessibility of the polyphenols compared to simply dispersing the polyphenol crystals in water. However, the chemical stability of the polyphenols during gastrointestinal passage was better for the crystalline form, which may have been because fewer of the polyphenol molecules were directly exposed to stressors in the aqueous phase. This study shows that pH-driven approaches are effective in introducing multiple polyphenols into plant-based milk samples. Moreover, the encapsulated polyphenols have a relatively high bioaccessibility, which may be advantageous for their use as nutraceuticals in functional foods and beverages foods.

## Figures and Tables

**Figure 1 foods-12-03432-f001:**
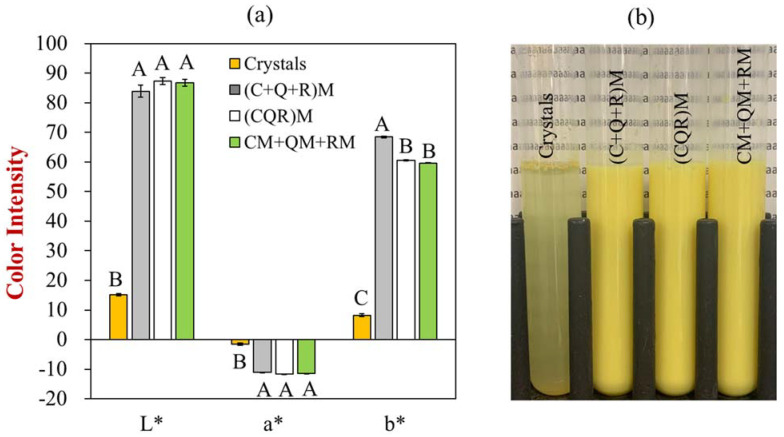
(**a**) Tristimulus color values (L*, a*, and b*) and (**b**) overall appearance of soymilk containing multiple polyphenols. Here, L* indicates lightness, a* is the red/green coordinate, and b* is the yellow/blue coordinate. Different letters (A, B, C) represent significant differences (*p* < 0.05).

**Figure 2 foods-12-03432-f002:**
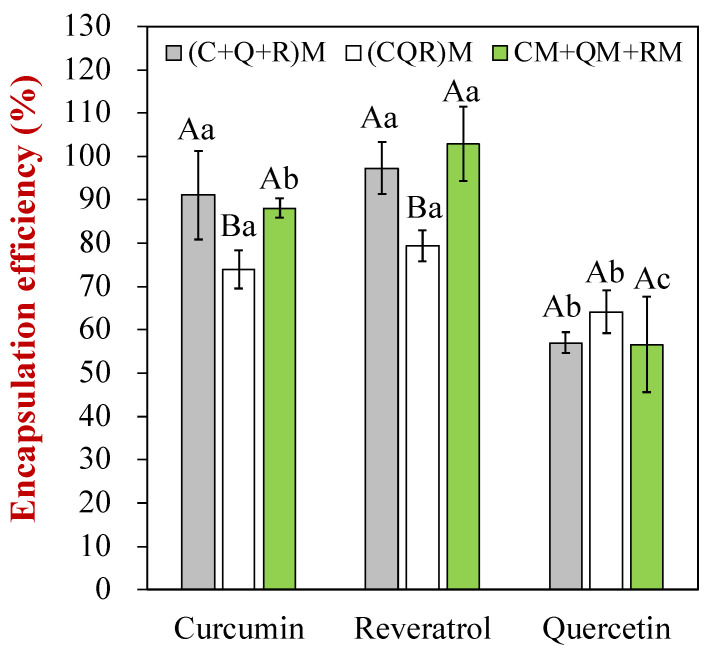
Comparisons of the encapsulation efficiency of each polyphenol for three pH-driven encapsulating protocols. Different letters (A, B, C, a, b, c) represent significant differences (*p* < 0.05) within or between polyphenol classes, respectively.

**Figure 3 foods-12-03432-f003:**
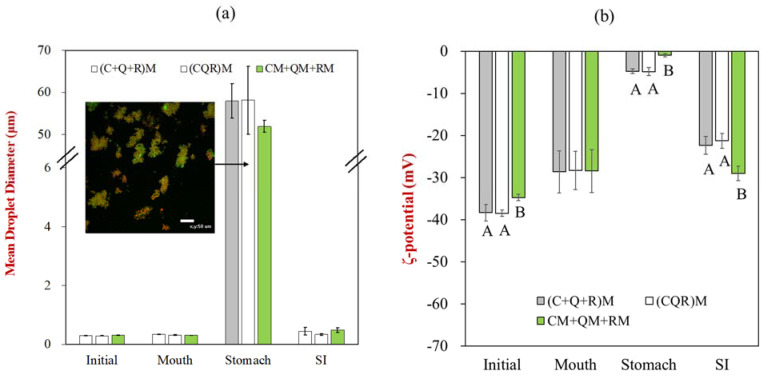
(**a**) The mean particle diameters and (**b**) zeta-potentials of multiple polyphenol-encapsulated soymilks under different gastrointestinal phases for samples prepared using three different pH-driven protocols. The image inserted is the confocal image at the stomach phase (scale bar 50 μm). The upper-case letters (A, B, C) represent significant differences between samples. The data without letters were not significantly different.

**Figure 4 foods-12-03432-f004:**
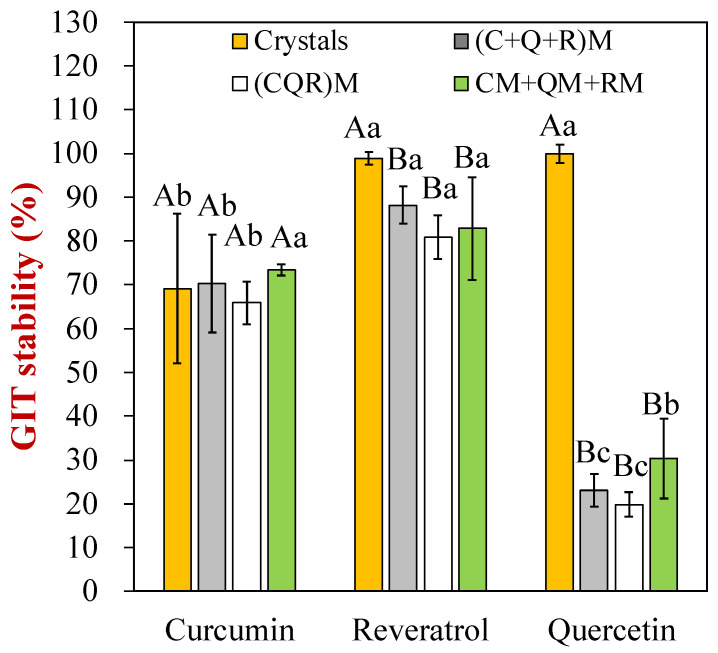
The gastrointestinal stability of each polyphenol (curcumin, resveratrol, and quercetin) under four different encapsulation protocols. Different letters (A, B, C, a, b, c) represent significant differences (*p* < 0.05) within or between polyphenol classes, respectively.

**Figure 5 foods-12-03432-f005:**
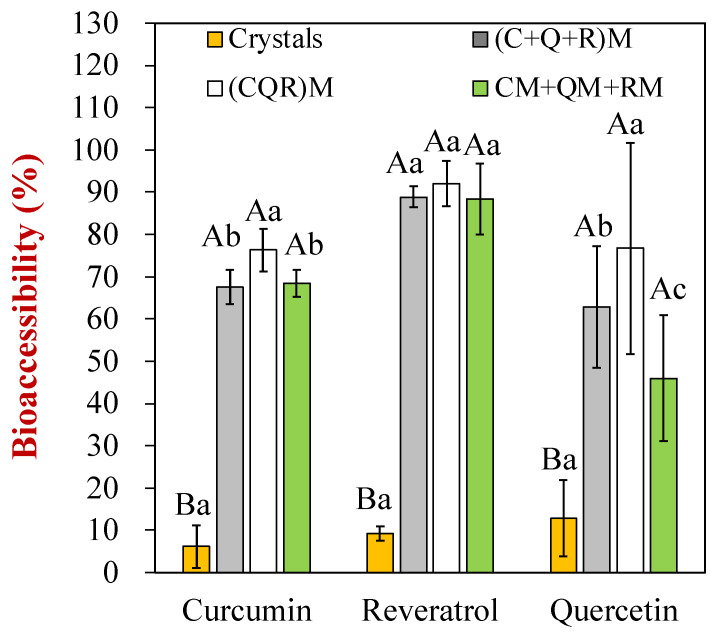
The bioaccessibility of each polyphenol (curcumin, resveratrol, and quercetin) for four different encapsulation protocols. Different letters (A, B, C, a, b, c) represent significant differences (*p* < 0.05) within or between polyphenol classes, respectively.

## Data Availability

The data that support the findings of this study are available from the corresponding author upon reasonable request.

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
