# Peer review of "Co-Encapsulation of Multiple Polyphenols in Plant-Based Milks: Formulation, Gastrointestinal Stability, and Bioaccessibility"

_foods, 2023, doi:10.3390/foods12183432_

Round 1
Reviewer 1 Report
Dear Authors,
The Article entitled “Co-encapsulation of Multiple Polyphenols in Plant-Based Milks: Formulation, Gastrointestinal Stability, and Bioaccessibility” is really an interesting one but very badly designed. The fallowing is few suggestions for a bit betterment of the article
1. The introduction of the article is very informative but the corelation of the introduction with the aim of the study is not clear. Authors should clear the reason for the selection of the three different formulation protocols
2. Which alkaline solution used in the study. Authors should add the pH of the alkaline solution in the method section.
3. What is the reason for the selection of the soymilk over other plant-based milks?
4. Mention the mouth pH in the Mouth Phase section.
5. Is the HPLC method for the polyphenol estimation is a simultaneous method or authors estimate the component separately. If it’s a simultaneous methods, authors should provide the method development plots for the HPLC study.
6. Authors should clarify the L*, a*, b* values use in colour
7. The Figure 1a “ Color” can be change to colour intensity to justify the numbers.
8. The figure 2 , 4 and 5 is confusing. What do the authors wants to show in the X axis of the graph? What are A, B and C as well as a,b,c ?
9. The resolution of Figure a (insert) is really low. Please provide high-resolution pictures.
A minor check of the English grammar is needed.
Reviewer 2 Report
In this study, three pH-driven approaches were used for fabricating polyphenols-fortified soymilk. The effect of three pH-driven approaches on co-encapsulated efficiency, GIT stability, and bioaccessibility of three bioactive polyphenols (curcumin, resveratrol, and quercetin) into soymilk was investigated. The topic is interesting and may provide beneficial information for the preparation of polyphenol-fortified plant-based beverage. Some minor comments are as follows:
Line 63. Please, deleted “EDITED TO HERE”.
Lines 64-74. Please, replaced “;” with “,”.
Lines 87-88. Why did the authors select curcumin, resveratrol, and quercetin as model polyphenols, and on what basis were they selected?
Lines 145 and 148. Reference format is wrong. Please modify.
Lines 229-233. Why does it take longer for dissolving three polyphenols into one alkaline solution?
Lines 286-288. The real intestinal environment is relatively closed. Is quercetin more stable when digested in a closed environment?
Lines 301-315. With 70% of quercetin already degraded during digestion, is it reasonable to use the remaining quercetin to calculate bioaccessibility?
In this study, three pH-driven approaches were used for fabricating polyphenols-fortified soymilk. The effect of three pH-driven approaches on co-encapsulated efficiency, GIT stability, and bioaccessibility of three bioactive polyphenols (curcumin, resveratrol, and quercetin) into soymilk was investigated. The topic is interesting and may provide beneficial information for the preparation of polyphenol-fortified plant-based beverage. Some minor comments are as follows:
Line 63. Please, deleted “EDITED TO HERE”.
Lines 64-74. Please, replaced “;” with “,”.
Lines 87-88. Why did the authors select curcumin, resveratrol, and quercetin as model polyphenols, and on what basis were they selected?
Lines 145 and 148. Reference format is wrong. Please modify.
Lines 229-233. Why does it take longer for dissolving three polyphenols into one alkaline solution?
Lines 286-288. The real intestinal environment is relatively closed. Is quercetin more stable when digested in a closed environment?
Lines 301-315. With 70% of quercetin already degraded during digestion, is it reasonable to use the remaining quercetin to calculate bioaccessibility?
Reviewer 3 Report
The present work is based on the encapsulation of three polyphenols using the pH shift method. It is well justified and structured and is presented as a continuation of previous work since they evaluate the encapsulation of three polyphenols at the same time and using three different methods.
I found a lack of description in material and method section. It is not clear how they calculate the amount of polyphenols that are actually encapsulated. I also found necessary to detailed how many of polyphenols are in the surface of this particles.
In formulas 1, 2 and 3 it must be an error since there is two different definitions for CM.
Furthermore, it is not clear how authors calculate the GIT stability and then bioaccessibility. Both parameters should be related to the initial concentration of polyphenols, if it is so, it is not possible that for example the 20-30% of quercetin resist the gastrointestinal digestion, while having a 60% of bioaccesibility. Both percentages should be expressed to the initial concentration to be valid and to drive proper conclusions. Please modified and clarify methology and calculations.
Round 2
Reviewer 1 Report
Dear Authors
The article is revised and good to be accepted in the Journal
Minor spelling checks are needed